# An Artificial Neural Network for Solar Energy Prediction and Control Using Jaya-SMC

Mokhtar Jlidi [1], Faiçal Hamidi [1,*], Oscar Barambones [2,*], Rabeh Abbassi [3], Houssem Jerbi [4], Mohamed Aoun [1] and Ali Karami-Mollaee [5]

[1] Laboratory Modélisation, Analyse et Commande des Systèmes, University of Gabes, Gabes LR16ES22, Tunisia
[2] Automatic Control and System Engineering Department, University of the Basque Country, UPV/EHU, Nieves Cano 12, 01006 Vitoria-Gasteiz, Spain
[3] Department of Electrical Engineering, College of Engineering, University of Ha'il, Hail 1234, Saudi Arabia
[4] Department of Industrial Engineering, College of Engineering, University of Ha'il, Hail 1234, Saudi Arabia
[5] Faculty of Electrical and Computer Engineering, Hakim Sabzevari University, Sabzevar 96186-76115, Iran
[*] Correspondence: faicalhamidi@yahoo.fr (F.H.); oscar.barambones@ehu.eus (O.B.); Tel.: +34-94-501-3235 (O.B.); Fax: +34-94-501-3270 (O.B.)

**Abstract:** In recent years, researchers have focused on improving the efficiency of photovoltaic systems, as they have an extremely low efficiency compared to fossil fuels. An obvious issue associated with photovoltaic systems (PVS) is the interruption of power generation caused by changes in solar radiation and temperature. As a means of improving the energy efficiency performance of such a system, it is necessary to predict the meteorological conditions that affect PV modules. As part of the proposed research, artificial neural networks (ANNs) will be used for the purpose of predicting the PV system's current and voltage by predicting the PV system's operating temperature and radiation, as well as using JAYA-SMC hybrid control in the search for the MPP and duty cycle single-ended primary-inductor converter (SEPIC) that supplies a DC motor. Data sets of size 60538 were used to predict temperature and solar radiation. The data set had been collected from the Department of Systems Engineering and Automation at the Vitoria School of Engineering of the University of the Basque Country. Analyses and numerical simulations showed that the technique was highly effective. In combination with JAYA-SMC hybrid control, the proposed method enabled an accurate estimation of maximum power and robustness with reasonable generality and accuracy (regression (R) = 0.971, mean squared error (MSE) = 0.003). Consequently, this study provides support for energy monitoring and control.

**Keywords:** JAYA algorithm; forecasting; artificial neural networks; sliding mode control; PEMFC; MPPT; SEPIC chopper

## 1. Introduction

Nowadays, the prediction of the developments that will occur in all areas is becoming a critical necessity. Therefore, the techniques that have merged to predict a future result or ensure the so-called forecasting are increasingly being demanded. Among such techniques, the artificial neural network (ANN) has been used as a forecasting tool in many applications. Indeed, it has been used in the tourism sector to forecast the number of tourists or hotel stays in a country [1]. It has also been used in the financial trading [2] and energy sector, including power usage and especially renewable energy [3–5]. The literature has shown many published studies highlighting the performance of neural network-based methods compared to other forecasting methods, particularly with high-frequency data. The present investigation focuses onrenewable energies application because it is among the most important fields attracting the researchers worldwide. This is justified by the fact that such energies are sustainable and preserving the ecosystem. Solar energy and wind energy are the most important and widely used renewable energy sources. Solar energy resources are

considered an efficient source of sustainable energy. Photovoltaic solar energy is converted into electricity using photovoltaic cells. Today, the energy extracted from photovoltaic (PV) power plants can be off-grid or integrated into the electrical grid [6], especially when they are connected to other energy sources to form a hybrid system [7,8]. As an innovative forecasting framework, FE-SVR-mFFO has been proposed as a hybrid technique for improving strategic decision making in smart grid applications [9]. Based on the FE-SVR-mFFO results, it has been demonstrated that it is effective in terms of stability, accuracy, and convergence rate.

To achieve the sizing of the photovoltaic generator, its produced energy under continuous weather changes should be estimated. Consequently, weather changes should be predicted in order to facilitate the PV system control and monitoring. There are two types of forecasting methods: the qualitative methods based on sensory emotions, doubts, experiences, and opinions; and the quantitative methods depending entirely on mathematical calculations. In relation to the studied application of PV systems, the second method is the most commonly used method to predict the temperature and the solar irradiation.

Despite the advantages of solar energy, photovoltaic generators have some drawbacks, such as a lower efficiency compared to fossil energy sources, as well asan instability of the produced energy since it is influenced by the changes in the atmospheric conditions. This issue seriously affects the grid stability. The energy produced by a photovoltaic generator changes with sunlight and temperature and can also provide a better energy conversion at a specified operating point. This point is called the maximum power point (MPP). The researchers designed several techniques for tracking the MPP to achieve voltage and current regulation [10]. It has been mentioned in [11,12] that the maximum power point tracking (MPPT) control techniques based on electric current measurement are used; however, they have some drawbacks related to employed analogue controllers. However, digital MPPT controllers are designed by algorithms, as investigated in [7,11,13,14]. In the literature, many methods to search the MPP point have been proposed. The most famous one is the perturb and observe (P&O) algorithm, which is the oldest and simplest one. In [15], it has been indicated that the algorithm (P&O) is generally dependent on initial conditions and exhibits fluctuations throughout the optimum point. Otherwise, this method has a bad performance in the case of sudden changes in meteorological conditions [16–20]. Taking that into consideration, another MPPT technique called incremental conductance (IC) has been proposed to overcome the limits of the P&O algorithm. The IC is considered more complex than the P&O [21]. The algorithms based on the measurement of the open circuit voltage (Voc) or the short circuit current (Isc) are veritably simple and easy to apply. Nevertheless, their main problem is the power loss and the transmission interruption during Voc and Isc measurement [22,23].

To overcome this problem, an experimental cell of the same nature as the PV panel cells is used. In addition, locating the optimal operating point is particularlychallenging. Therefore, these methods are only approximations that do not give enough precision, and therefore, the system does not necessarily operate at the optimal point. In this context, the technique based on fuzzy logic control (FLC) theory is important and effective [24]. Indeed, this technique works at the optimal point without oscillations [25]. However, the implementation of the FLC technique is more complex than classical algorithms and its efficiency depends largely on the rule table.

In a more developed context, many optimization techniques, some based on the heuristic approach and others on the meta-heuristic approach, have been used for the implementation of MPPT techniques. Swarm-based algorithms (particle swarm, artificial bee colony,...), and trajectory-based algorithms (Tabu search, hill climbing,...), as well as evolutionary algorithms (genetic algorithm,...) have been suggested [26,27]. In particular, the evolutionary intelligence and swarm based algorithms are probabilistic algorithms that require common control parameters and are very efficient for MPP point tracking. Therefore, the choice of a reliable and consistent MPPT technique was targeted towards the JAYA algorithm [28]. This algorithm was formulated by RAO in 2016 to solve constrained

and unconstrained optimization problems, based on the concept that the solution obtained for a given problem should move towards the best solution and should avoid the worst solution [29]. The JAYA algorithm is a definitive, parameter-free algorithm.

In contrast to other meta-heuristic algorithms, the JAYA algorithm does not require any special parameters. Compared to, for instance, GA, PSO, CPSO, and GWO, each algorithm requires careful selection of parameters. It is because they affect the accuracy of the search for the optimal value. A further advantage of JAYA is that it has a very simple structure and has been shown to be effective at solving optimization problems [30,31]. Since SMC has a simple algorithm and a high degree of robustness, it has been widely used for nonlinear control systems [32]. As a result of the simplicity of both methods, as well as their effectiveness, JAYA and SMC have been combined in this work. It has been shown in [33] that gradient optimization techniques are combined with the PI controller to determine the optimal point and control for SEPIC. However, due to the complexity of this controller, it is difficult to extract the PI gains. Furthermore, the gradient optimization method requires special parameters to reduce error, which are difficult to determine. In this way, hybridization between JAYA and SMC provides a simple and robust means for controlling and determining the optimal point.

In this paper, the JAYA algorithm has been exploited to control the DC-DC converter and ensure MPP point tracking. To control the duty cycle, there are many control methods. Among the best-known methods are those that use PID or PI controllers [33,34]. These methods are simple and easy to use as their parameters (P proportional, D derivative, P integral) are relatively easy to measure. However, these methods are not practical in industrial or energy environments as they have many non-constant inputs that change continuously with the weather and external factors. As a result, these methods provide limited performance in the presence of external disturbances and uncertainties. On the other hand, sliding mode control (SMC) is an important technology in many dynamic and complex domains, mainly because of its significant ability to reject uncertainties and exclude external disturbances. SMC has been widely used for non-linear control systems due to its simple algorithm and high robustness.

By predicting the operating temperature and radiation of the PVS, the proposed research makes use of artificial neural network (ANN) technology to predict the current and voltage. Further, the JAYA-SMC hybrid control system is designed to determine the MPP and duty cycle of the single-ended primary-inductor converter (SEPIC) that supplies the DC motor.

Even with the constant evolution of photovoltaic panel technology in the field of electricity generation, the storage of electrical energy for later use or to reduce the fluctuation of solar production remains a great challenge for users of this technology when not supplied by the grid. Possible solutions include batteries for storage, which are expensive and difficult to maintain safely and need to be replaced frequently. To this end, other new and more viable technologies for energy storage, such as systems that rely on hydrogen storage, have become available due to their significant advantages over battery systems [35–37]. The underlying technology that makes this possible is hydrogen electrolysis, which is the process where the use of low voltages is used to create reactions in various solutions. This is the electrolytic process, which can be used to split a molecule of water into hydrogen and oxygen [38].

Energy is of the utmost importance, as mentioned above, as such it is imperative to develop management methods, the most notable of which are forecasting techniques and MPPT detection. Recent years have seen the development of these methods. In this respect, choosing the most appropriate MPPT method and making a prediction can be a challenge, since each method has its own advantages and disadvantages. In light of these factors, it is imperative to select simple, effective, and straightforward methods, particularly when there are unpredictable weather conditions that can prevent some methods from performing properly, e.g., GA and PSO. For control and monitoring, critical evaluation and analysis are key. A specific idea presented in this paper is combining robust control and simple

optimization techniques for energy prediction and control, while maintaining robustness and accuracy.

This paper makes the following major contributions:

- Implementation of artificial neural networks (ANN) to predict temperature and solar radiation as it is one of the most effective and efficient methods in all fields.
- Implementation of JAYA-SMC based approach to control DC-DC converters according to the maximum power point tracking concept (MPPT).

The remainder of the paper is organized as follows: In the second section, the methodology is discussed, including PV panel modeling, Artificial Neural Network modeling, and the implementation of JAYA-SMC hybrid controllers for MPP extraction and SEPIC control. Detailed simulation results and discussions are presented in Section 3. This paper ends with concluding statements and recommendations for future work.

## 2. Methodology

The topology of the studied system is depicted in Figure 1.

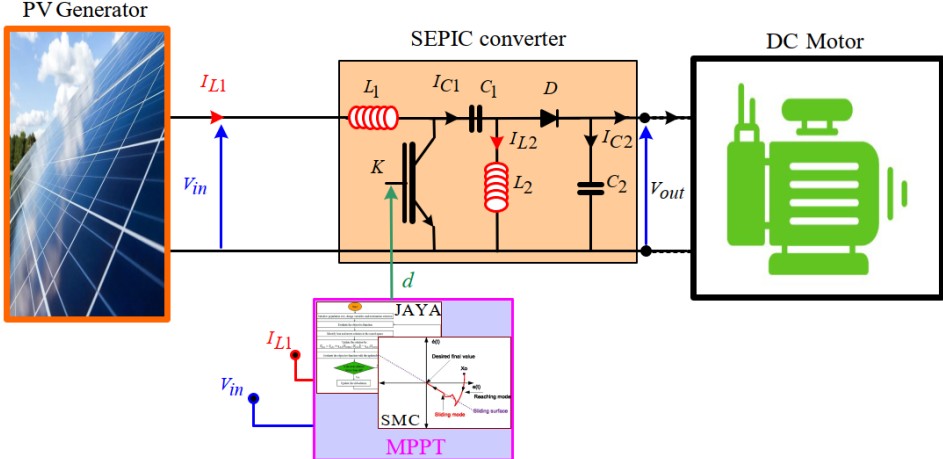

**Figure 1.** Topology of the studied system.

### 2.1. PV Panel Modeling

The solar cell is a PN junction semiconductor. When exposed to light, it generates a direct electric current. The generated current varies slightly linearly with solar radiation variation. The commonly used circuit model of a PV cell consists of a current source that depends on the values of solar radiation and temperature in parallel with a diode and a shunt resistor ($R_{sh}$), that are in series with a second resistor ($R_s$) (Figure 2).

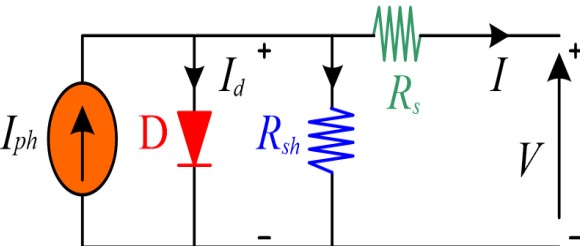

**Figure 2.** PV panel equivalent circuit.

According to Kirchhoff's current law, the generated current *I* is represented by Equation (1)

$$I = I_{ph} - I_0 (e^{\frac{q(V+IR_s)}{aKTN_s}} - 1) - \frac{V + IR_s}{R_{sh}} \tag{1}$$

where $V$ is the voltage at the PV generator terminals, q is the charge of an electron ($1.602 \times 10^{-19}$ C), a is the PV cells ideality factor, $N_S$ is the number of PV cells connected in series, $K$ is the Boltzmann's constant ($1.38 \times 10^{-23}$ J/K), $N_S$ is the number of series cells, $I_{ph}$ is the photocurrent, described by Equation (2), and $I_0$ is the saturation current of the diode expressed by (3).

$$I_{ph} = G \frac{I_{SC} + K_i(T - T_{STC})}{G_{STC}} \tag{2}$$

where $G_{STC}$ and $T_{STC}$ are, respectively, the irradiation and temperature under standard test conditions, $G$ and $T$ are their reel values, $I_{SC}$ is the short-circuit current, and $K_i$ is the temperature coefficient for short-circuit current.

$$I_0 = \frac{I_{SC} + K_i(T - T_{STC})}{e^{\frac{q[V_{OC} + K_v(T - T_{STC})]}{aKTN_S}} - 1} \tag{3}$$

where $V_{oc}$ is the open-circuit voltage and $K_V$ is the temperature coefficient for open circuit voltage.

As can be seen in Equation (2), the photocurrent depends mainly on meteorological conditions such as irradiation and temperature. The mathematical development conducted in the majority of literature confirms that the irradiance has more effect on the photocurrent and consequently on the generated current.

### 2.2. Proposed Artificial Neuro Networks Predictive Modeling

Forecasting allows for predicting the required future steps for different application areas, such as commercial and industrial fields. This is mandatory for to make the necessary decisions to improve the future situation and avoid the worst outcomes that could affect the desired situation. Forecasting is divided into two types: qualitative and quantitative forecasting. Time series forecasting is the most prominent of these methods. Indeed, time series are predicted by mathematical forecasting through time-specific historical data. The historical data are analyzed and strategic decision-making is performed for the future. For this type of prediction, the analysis must be thorough and evidence-based to ensure that the future outcome is achievable.

Nowadays, most technologies use artificial intelligence (AI) because of its efficiency. Among AI techniques, those based on neural networks (NN) are being employed to address many problems, including prediction problems. The working principle of NNs is based on an interconnected processing element that depends on biological neurons equivalent to pieces that carry information and transmit it to other cells in a series of networks [39].

According to Figure 3, the artificial neurons consist of three levels: the first level is the input, which consists of a number of nodes where each node represents one of the inputs, and the second level is the hidden level, and its number varies from one network to another according to the level of input and output. The last level is the output level, which is the result or the goal to be reached. All the previous levels are connected to each other through the nodes and contain a group of nodes that receive inputs and outputs called the level, and each node carries weights that enhance the strength of the neural connection.

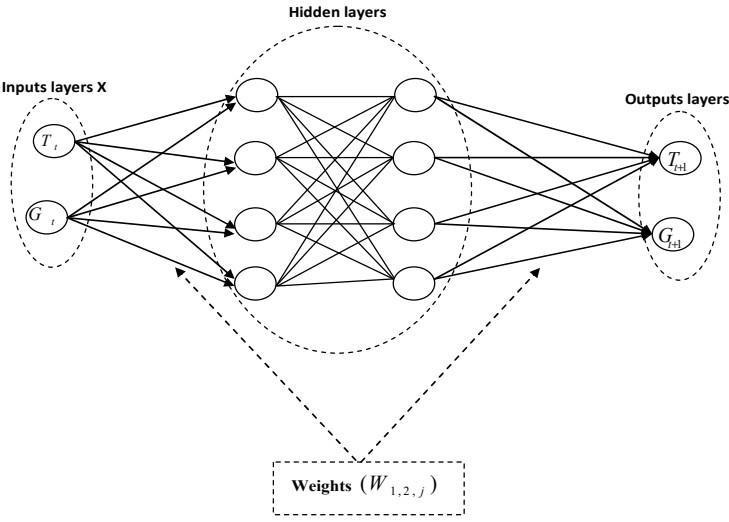

**Figure 3.** Neural network architecture.

A neuron consists of an integrator that performs the weighted sum of its inputs [40]. The result, *P*, of this sum is then transformed by a transfer function, Y, which produces the output, S, of the neuron, according to the Equations (4) to (8).

$$P = \sum_{j=1}^{n} W_{1,j} X_j - \beta \tag{4}$$

$$P = W^T X - \beta \tag{5}$$

$$X = [X_1, X_2, \ldots, X_k]; W = [W_{1,1}, W_{1,2}, \ldots, W_{1,n}] \tag{6}$$

$$W = \begin{bmatrix} W_{1,1} & W_{1,2} & \ldots & W_{1,n} \\ W_{2,1} & W_{2,2} & \ldots & W_{2,n} \\ W_{k,1} & W_{k,2} & \ldots & W_{k,n} \end{bmatrix} \tag{7}$$

To obtain the output of the neuron, the activation function Y is employed:

$$S = Y(P) = Y(W^T X - \beta) \tag{8}$$

where
X: input layers.
W: weights.
n: numbers of entries, for our example, n=2.
k: number of neurons in the same layer.
Y: activation equation.
$\beta$: bias.

Various possible forms of the transfer function can be found. There are four popular activation functions: symmetric threshold, threshold, sigmoid, and linear.

For the present case, the sigmoid function is used to forecast the outputs (T, G), according to the following input/output relationship (9):

$$S = \frac{1}{1 + e^{-(W*X - \beta)}} \tag{9}$$

The NN-based prediction technique of the temperature and the irradiation (T and G) follows these steps:

- Step1: Data assembly, pre-processing, data conversion, and normalization. The data set used to predict the temperature and solar radiation reflected on the PV under study was obtained from the Department of Systems Engineering and Automation at the

Vitoria School of Engineering of the University of the Basque Country. The data was collected using the irradiance and temperature sensor Si-V-010-T [41].

- Step2: Statistical analysis.
- Step3: Neural Network objects design.

The Figure 4 shows the structure of the adopted neural network.

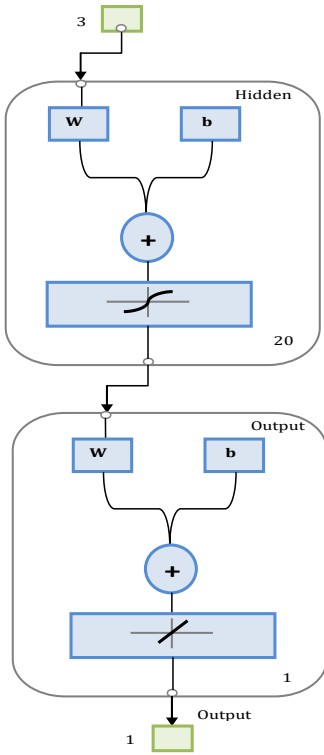

**Figure 4.** The neural network architecture.

- Step4: Network training; the algorithm of Levenberg Marquardt has been used for the training of the network. This choice has been justified by the fact that this algorithm typically requires more memory but less time. The training automatically stops when the generalization stops improving, as indicated by an increase in the mean square error of the validation samples. The Mean Squared Error is the average squared difference between outputs and targets. Lower values are better, as zero means no error. This algorithm is also improving the regression, *R*, and it is the value measuring the correlation between outputs and targets. A unit, *R*, value indicates a close relationship, while 0 denotes a random relationship.
- Step5: Simulation of network response to new entries.
- Step6: Approval and testing.

The sample data process is divided into three phases: A reasonable result can be achieved by adjusting the ANN weights during the training phase. The second phase involves determining the minimum point of error. In the third phase, the accuracy of the ANN is evaluated.

After presenting the proposed forecasting technique, the following section focuses on the suggested MPPT-based control of the DC-DC chopper integrated to extract the maximum available power independently of the meteorological conditions.

The main stages of the temperature and solar radiation prediction algorithm are displayed in a flowchart in Figure 5.

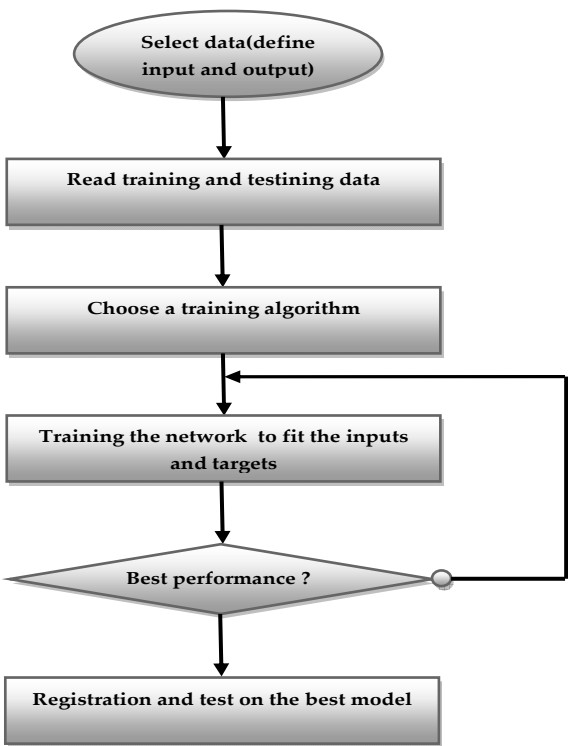

**Figure 5.** Flowchart of the temperature and solar irradiation prediction algorithm.

Hyperparameters have a significant impact on the neural network accuracy. For the purpose of determining the optimal set of hyper-parameters, Bayesian surrogate gaussian processes, gradient boost regression trees, and random forest were used [42]. The authors in [43] presented a meta-heuristic algorithm for optimizing hyperparameters. A cross-validation approach was used in this study to train and validate neural networks with multiple architectures. In this method, weight optimization was performed by finding the weights that minimized the mean squared errors (MSE) between the obtained outputs and the desired ones, using a set of training models, and comparing the results at each epoch against a different set of training models. In the second set, which was generally referred to as the validation set, training was stopped once the MSE in the validation set increases. As part of a batch training process, networks were trained using the back-propagation algorithm with different iterations or epochs. The number of epochs and the number of neurons in the hidden layer had a significant impact on the performance of the neural network. Using the hidden layer neural network of two neurons and training it for 1000 epochs, the smallest MSE validation could be obtained. In this study, 60538 temperature and solar radiation values were collected, of which 70% were recorded for training and 30% were used for validation and testing. With 20% of the hidden layer's size, this was achieved.

In this network, two layers of feed-forward neurons with sigmoid hidden neurons and linear output neurons were used to perform regression tasks. The neural network was trained using the Levenberg-Marquardt algorithm, in which training was automatically stopped when there was no further improvement in generalization. An analysis of the mean square error (MSE) and regression (R) indicated the completeness of this algorithm. According to the results of this study, MSE = 0.003 and R = 0.998 were found, indicating a strong correlation between outputs and objectives.

Concerning the choice of input features, various research works have dealt with this subject [44]. In the present work, each additional neuron allows for the consideration of specific profiles of the input neurons. A larger number, therefore, allows for it to be possible to better adhere to the data presented but reduces the generalization capacity of the network. At this time, there is no general rule but rules of thumb. A subsequent

research path would consist of estimating a network comprising of many neurons and then in simplifying it by analyzing multi-collinearities, by learning rule eliminating useless neurons, or by defining an architecture considering the structure of the variables identified beforehand by a principal component analysis.

### 2.3. JAYA-SMC Hybrid MPPT Control of the SEPIC Chopper

#### 2.3.1. Integrated SEPIC Chopper

The single-ended primary-inductor converter (SEPIC) is a modification of the Basic Boost and Cuck Converter. It is more performant than other DC-DC converters in terms of purity and efficiency of the input current since it shows very little bypass or ringing, as well as a reduced switching loss. The output noise and power phase that can be operated at a much higher frequency than that of other inverters will also be reduced. The output voltage achieved by the SEPIC converter is non-inverting. Figure 6 shows the electrical diagram of a SEPIC converter. The different values of the electronic components are mentioned in Appendix B.

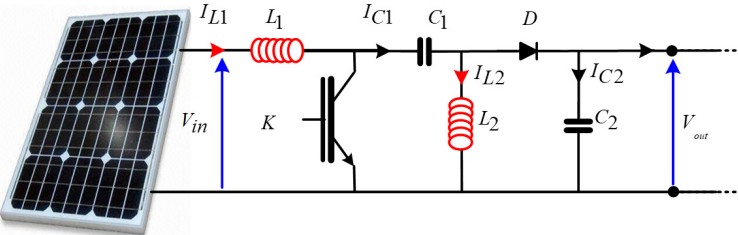

**Figure 6.** Electrical diagram of SEPIC converter.

The SEPIC chopper operation principle is analyzed in two stages according to the conduction state of the switch K. Based on the notations of the Figure 6, when the switch K is closed, the behavior of the SEPIC is described by Equation (10). When it is open, Equation (11) describe the behavior of the SEPIC.

$$\begin{cases} \frac{dil1}{dt} = \frac{1}{L_1} \cdot V_e \\ \frac{dil2}{dt} = \frac{1}{L_2} \cdot V_{c_1} \\ \frac{dV_{c_1}}{dt} = \frac{1}{C_1} \cdot i_{L1} \\ \frac{dV_{c_2}}{dt} = -\frac{1}{C_2} \cdot V_{c_2} \end{cases} \tag{10}$$

$$\begin{cases} \frac{dil1}{dt} = \frac{1}{L_1} \cdot V_e - \frac{1}{L1} \cdot [V_{C_1} + V_{C2}] \\ \frac{dil2}{dt} = -\frac{1}{L_2} \cdot V_{C2} \\ \frac{dV_{C_1}}{dt} = -\frac{1}{C_1} \cdot i_{L2} \\ \frac{dV_{C_2}}{dt} = \frac{1}{C_2} \cdot (il_1 + il_2) - \frac{1}{C_2} \cdot V_{C_2} \end{cases} \tag{11}$$

The state model of the converter is inferred from Equations (10) and (11). This model is presented in Equation (12).

$$\begin{cases} \frac{dil1}{dt} = \frac{1}{L_1} \cdot V_e - \frac{1}{L1} \cdot (1-d) \cdot [V_{c_1} + V_{c_2}] \\ \frac{dil2}{dt} = \frac{d}{L_2} \cdot Vc_1 - \frac{(1-d)}{L_2} \cdot V_{c2} \\ \frac{dV_{c_1}}{dt} = \frac{(1-d)}{C_1} \cdot il_1 - \frac{d}{C_1} \cdot i_{L2} \\ \frac{dV_{c_2}}{dt} = \frac{(1-d)}{C_2} \cdot (il_1 + il_2) - \frac{1}{C_2} \cdot V_{c_2} \end{cases} \tag{12}$$

The parameter *d* refers to the duty cycle of the SEPIC chopper. Such a parameter is mainly determined via the maximum power point tracking technique.

### 2.3.2. JAYA-SMC Hybrid MPPT Control

One of the most important problems that the PV generation faces is the problem of maximum power tracking ability, which is mandatory for the DC-DC chopper. In fact, the solution exists to control such a converter to take maximum advantage of the power produced by the PV panels. In order to overcome the drawbacks of the conventional MPPT methods, such as perturb and observe (P&O), incremental conductance (IC), etc. ..., a recent MPPT based on the hybridization of the JAYA optimization algorithm and the sliding mode control (SMC) is proposed. This MPPT technique is employed to maximize the power generated under the constraints of equality that includes the relationship between current and voltage. This problem aims to raise the PV power by setting the optimum value of *I* and *V*, where the equations expressing the optimization problem and the constraint of equality $R\,(I, V) = 0$ are established in Equations (13) and (14):

$$\begin{cases} \max\limits_{I,V}[P(I,V)] \\ R(I,V) = 0 \end{cases} \tag{13}$$

With

$$\begin{cases} P(I,V) = I \cdot V \\ R(I,V) = I_{ph} - I_0(e^{\frac{q(V+IR_s)}{aKTN_s}} - 1) - \frac{V+IR_s}{R_{sh}} - I \end{cases} \tag{14}$$

The objective is to solve the optimization problem based on equation in (13).

1.　Jaya Method

Heuristic algorithms are becoming a good solution to model-free optimization problems [45–47]. In this article, an advanced swarm-based JAYA algorithm is used to find the maximum power point for PV arrays. This algorithm does not require any specific parameters to be configured. Therefore, it is easy to implement without having to modify the initialization parameters.

In the present work, f (x) is considered as the objective function to be maximized as the power P(X) = P (V, I). The JAYA algorithm searches $V_{max}$ and $I_{max}$ allowing it to reach $P_{max}$. It is easy to implement, in each iteration of i, as it is assumed that there is a number of design variables, m, (j = 1, 2, m), and n, the number of candidate solutions that determine the community size, K, (k = 1, 2, n). The best candidate obtains the best value of P(X) in the candidate solution set and the worst candidate obtains the worst value of P(X) in the set of candidate solutions. If X(i,j,k) is the value of the variable, $j^{th}$, of the filter Kth, in iteration $i^{th}$ this value is adjusted in Equation (15).

$$X_n(i,j) = X(i,j,k) + r_{1,i,j}(X_{best}(j) - |X(i,j)|) - r_{2,i,j}(X_{worst}(j) - |X(i,j)|) \tag{15}$$

Herein, X(i,j) is jth from the assumption of the solution $i^{th}$, |X(i,j)| is the absolute value of X(i,j), Xbest(j) is the best solution, Xworst(j) is the worst solution, Xn(i,j) is the update of the variable X(i,j), and $r_{1,2}$ are two random numbers belonging to the interval [0,1].

The main steps of Algorithm 1 are illustrated in the following description.

---

**Algorithm 1:** JAYA Algorithm

---

Step 1: Set the population and the maximum number of iterations NPop and

Nmax.

Step 2: Determine the Xbest andXworst solutions.

Step 3: While gen <= $\leq$ Nmax

For I = 1 to Npop carry out:

$X_n(i,j) = X(i,j,k) + r_{1,i,j}(X_{best}(j) - |X(i,j)|) - r_{2,i,j}(X_{worst}(j) - |X(i,j)|)$

Obtain the update community and evaluate the new value, if the new value is

more suitable than the previous one, it will replace the old one.

End for; End while.

Step 4: Show existing solutions X(i) and f(X(i)).

---

It is important to recall that the JAYA algorithm is used to optimize the parameters of the sliding mode control.

2. Sliding Mode Control Technique

To determine the duty cycle of the SEPIC chopper, the sliding mode control (SMC) has been adopted. The SMC is a kind of variable structure control algorithm. The fundamental difference between sliding mode control and conventional control strategy is the discontinuity of its control [48]; that is, the controller output changes over time and exhibits switching characteristics. Since the sliding mode is independent of system parameters and disturbances, the sliding mode system has strong robustness. The basic principle of sliding mode is shown in Figure 7, where the following steps are processed:

- First, design a sliding surface in state space.
- Have a selection of a control law to force the state trajectory of the system to move towards a predetermined surface in finite time.
- Maintain around this surface with appropriate switching logic.

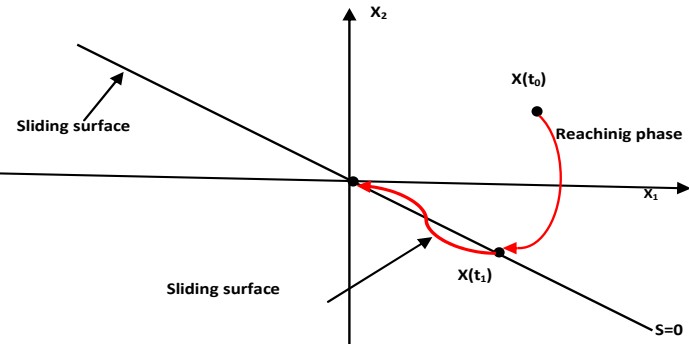

**Figure 7.** SMC basic principle.

In order to regulate the converter input current, the following sliding surface is chosen:

$$S = i_{ref} - i_{L1} \tag{16}$$

where $i_{ref}$ is the optimum current of the MPPT control and $i_{L1}$ the input current of the converter SEPIC. By defining the above surface, the control law should be applied to the converter to force the system to move on the sliding mode surface in a finite time, according to the following structure for control input:

$$U(t) = U_{eq}(t) + U_n(t) \tag{17}$$

where $U_{eq}$ defines the system's behavior on the sliding surface and is known as the equivalent control-input, $U_n$ is the nonlinear switching input that moves the state to the sliding surface and maintains the state on such surface in the presence of the uncertainty. $U_{eq}$ is obtained from the invariance condition and is presented as below: $(S = 0, \dot{S} = 0) \Leftrightarrow U = U_{eq}$

$$\dot{S} = \dot{i}_{ref} - \dot{i}_{L1} = 0$$
$$\dot{i}_{ref} - \frac{1}{L_1} \cdot V_e + \frac{1}{L1} \cdot (1 - U_{eq}) \cdot [V_{c_1} + V_{c_2}] = 0 \tag{18}$$

$$U_{eq} = \frac{V_{c_1} + V_{c_2} - V_e}{[V_{c_1} + V_{c_2}]} \tag{19}$$

Here, $U_n$ is chosen so that the Lyapunov stability criteria $(\dot{V} < 0)$ is met. Where

$$V = \frac{1}{2}S^2 \tag{20}$$

$$\dot{V} = S \cdot \dot{S} < 0$$
$$S \cdot \left( -\frac{1}{L_1} \cdot V_e + \frac{1}{L1} \cdot (1 - U) \cdot [V_{c_1} + V_{c_2}] \right) < 0 \tag{21}$$

$$S \cdot \left( -\frac{1}{L_1} \cdot V_e + \frac{1}{L1} \cdot (1 - U_{eq} - U_n) \cdot [V_{c_1} + V_{c_2}] \right) < 0 \tag{22}$$

$$-\frac{[V_{c_1} + V_{c_2}]}{L1} \cdot U_n \cdot S < 0$$
$$\Rightarrow S \cdot U_n > 0 \tag{23}$$

The nonlinear component is given by:

$$U_n = K. \, \text{sign}(S), \tag{24}$$

where $K > 0$; $\text{sign}(S) = \begin{cases} +1 \; si \; S > 0 \\ -1 \; si \; S < 0 \end{cases}$.

The gain $K$ is chosen to be positive. The choice of this gain is very influential since if it is small, the controller loses the robustness properties and if it is large, important oscillations will be derived at the level of the control unit. These oscillations can excite the neglected dynamics (Chattering phenomenon), or even damage the control unit [49]. Chattering can be reduced by replacing the "sign" function with a hyperbolic tangent function (tanh) [50]. The Equation (25) represents the new form of the command law:

$$U = \frac{1}{2}(1 + \tanh(s)) \tag{25}$$

## 3. Simulation Results and Discussion

The studied system depicted in Figure 1 integrates the PV generator, the DC-DC chopper, and the DC load. To simulate a more intricate consumer, the selected load was a DC motor. The Appendix A, Appendix B, and Appendix C summarize, respectively, the characteristics of the peimar SG340P commercial PV panel, the DC-DC converter, and the DC motor used to carry out the simulation.

The first simulations have been focused on the SG340P PV panel behavior when faced with different meteorological conditions of temperature and irradiance. In this context, Figures 8 and 9 represent the I-V and P-V characteristics of the adopted PV panel, under different temperatures and irradiations.

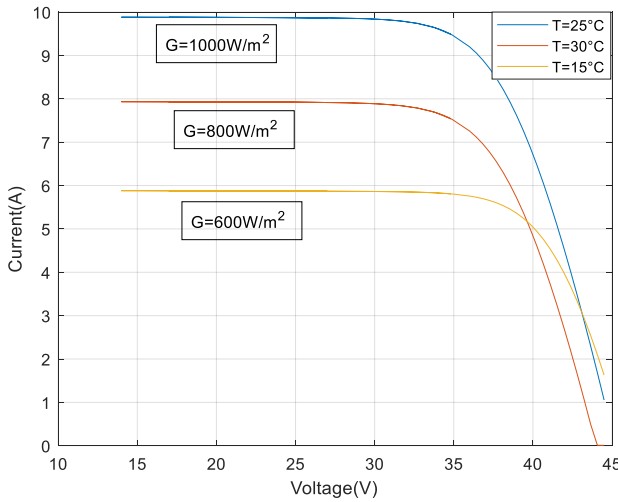

**Figure 8.** PV panel I-V characteristic.

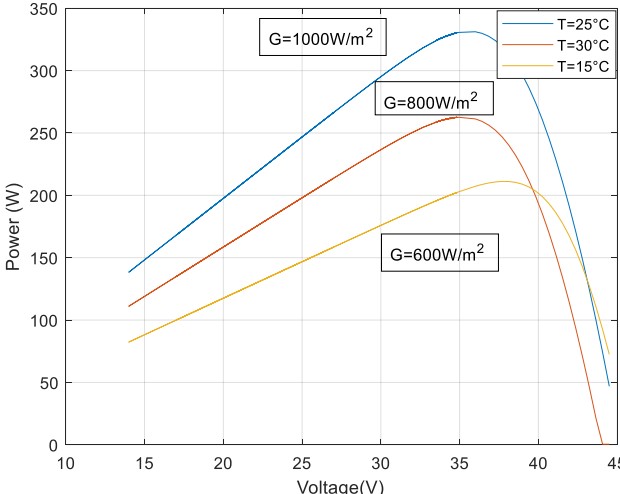

**Figure 9.** PV panel P-V characteristic.

To test the effectiveness of the JAYA algorithm, this method has been applied with variable temperature and solar radiation. Figure 10 shows the variable test conditions used.

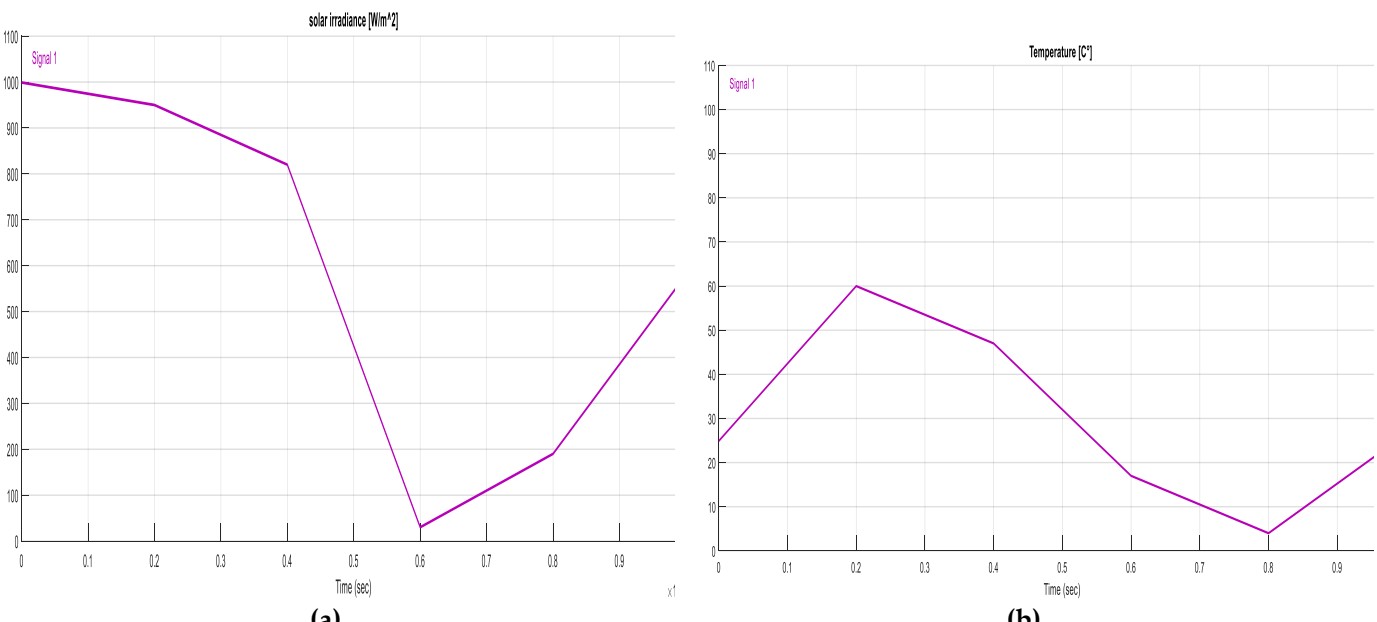

**Figure 10.** Solar irradiance and temperature variation: (**a**) Solar irradiance variation, (**b**) Temperature variation .

Figure 11 indicates that, in most cases, the power curve is smaller when compared to the maximum value. The power curve is, in most cases, smaller than the maximum value. Despite abrupt changes, the MPP detection algorithm performs satisfactorily. Due to the complexity of calculations, this is the case. Indeed, there is an inherent limitation to other methods in this regard.

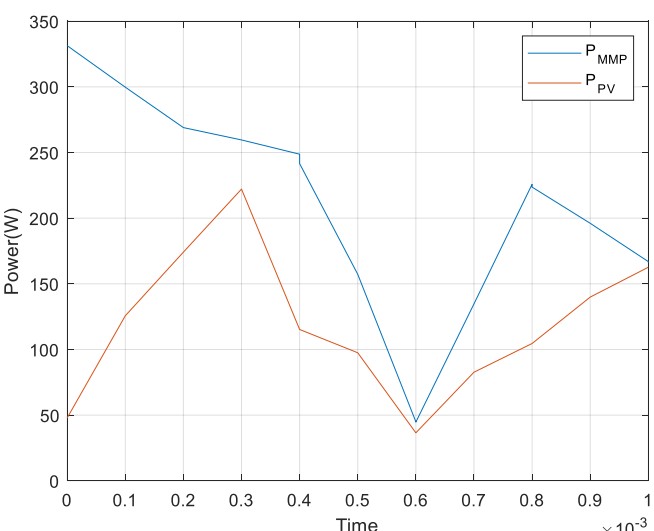

**Figure 11.** PV testing of the optimization algorithm.

At this stage, the main objective is to assess the performance and accuracy of the forecasting system. For the training, the errors were repeated 6 times after 200 epochs and the test was stopped at epoch 212 with a gradient of 0.09. The error is repeated from the 200th era, which showed the growing importance of data. Therefore, age 200 is chosen as the base and the weights are chosen as the final weights. In addition, the validation is 6 to 212 epochs, sincethe errors are repeated 6 times before the process stops definitively (see Figure 12).

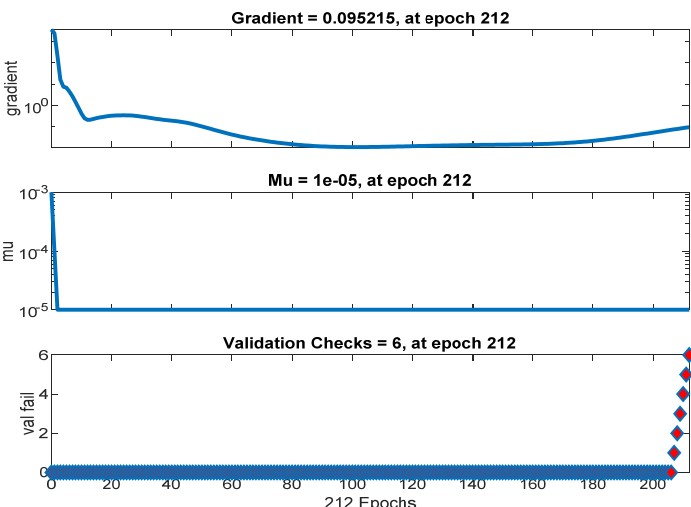

**Figure 12.** ANN training.

In Figure 13, the regression and its values measure the correlation between outputs and targets are presented. The *R* value is almost equal to 1, which means a close relationship.

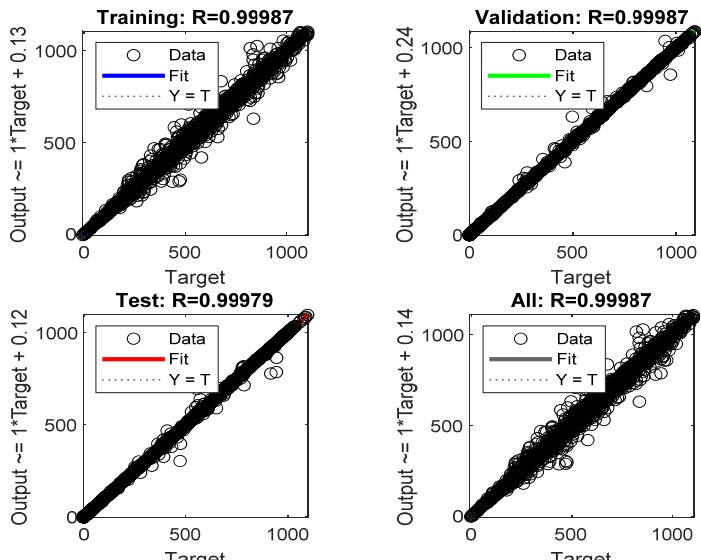

**Figure 13.** ANN regression.

Figure 14 illustrates that the MSE approaches 0 for the training and validation losses at epoch 10. The graph below illustrates the performance of the proposed method.

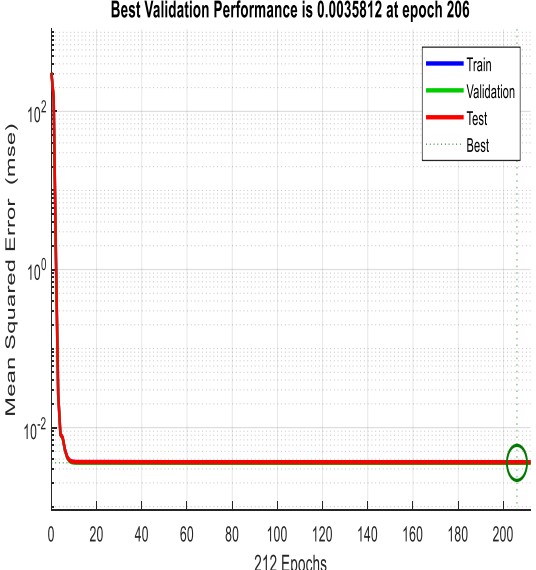

**Figure 14.** ANN performance.

Figure 15 represents the ANN forecasting model on Simulink.

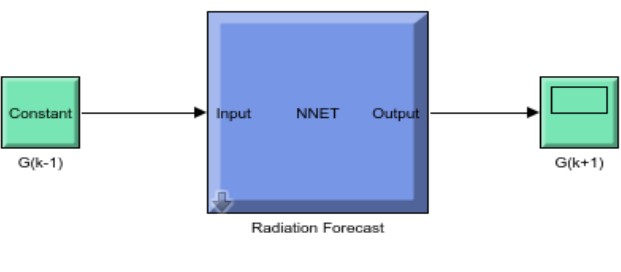

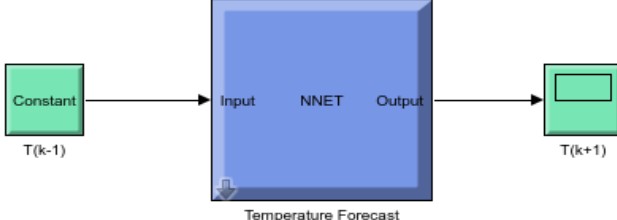

**Figure 15.** Simulink model of ANN forecasting.

According to the ANN forecasting model of Figure 14, the results found after the test of neurons are shown in Tables 1 and 2:

**Table 1.** Temperature forecast testing.

| Inputs | Real Temperature | Forecasted Temperature | Error |
|---|---|---|---|
| (54.43;54.50;54.54) | 54.59 | 54.58 | 0.01 |
| (26.15;26.15;26.06) | 25.96 | 26.00 | 0.04 |
| (59.86;59.76;59.91) | 60.08 | 60.09 | 0.01 |

**Table 2.** Irradiation forecast testing.

| Input | Real Irradiation | Forecasted Irradiation | Error |
|---|---|---|---|
| (959.28;960.38;961.85) | 962.58 | 962.71 | 0.13 |
| (548.03;547.66;546.75) | 545.65 | 546.1 | 0.45 |
| (877.25;877.80;877.80) | 876.70 | 877.38 | 0.68 |

-The inputs are the last three previous temperatures of the PV panels ($T_k$, $T_{k-1}$, and $T_{k-2}$).

-The real temperature is the temperature found in the real-time.

-The forecasted temperature is the forecasted temperature found by the neural network-based technique ($T_{k+1}$).

The proposed neural network-based forecasting technique is tested on the real data and comparisons between the real and the predicted temperature, as well as the irradiation are conducted with the aim of reducing the error between the target and the forecasted values.

The used MPPT has a sampling time of $10^{-2}$ seconds; depending on its output the sliding mode controller has been used to find the duty cycle.

According to the same principle of using the last previous three values of temperature and irradiance to deduce their forecasted values, Figure 16 represents the results of temperature and irradiation forecasting.

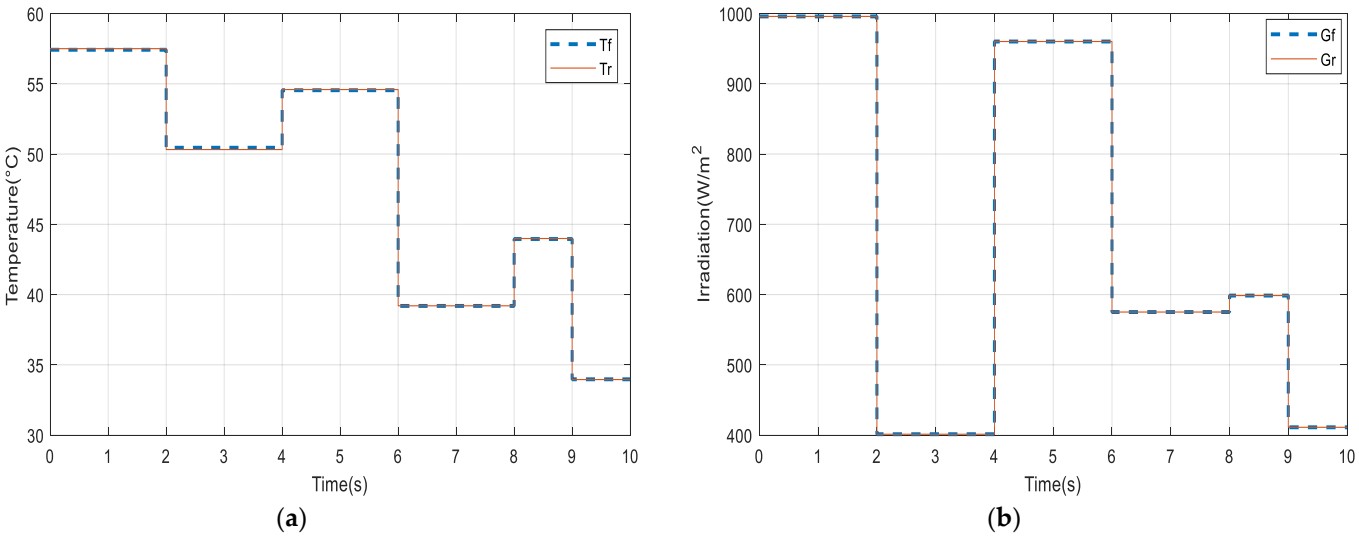

**Figure 16.** Forecasting results: (**a**) Temperature forecast, (**b**) Irradiation forecast.

From Figure 16, it can be observed that there is a good agreement between the expected values and the real ones. As can be seen, the SEPIC duty cycle is adjusted in accordance with the changes in the MPPT. Thus, it demonstrates its ability to adapt to changes in the system as they occur. As shown in Figure 17, the difference between the reference current ($I_{mpp}$) and the input current of the SEPIC is approximately 10%. It is pertinent to note that this error is very small, which indicates the accuracy of the SMC. According to the graph, there are some peaks caused by changes in the reference current. These changes occur within 25 milliseconds without causing any problems for the system.

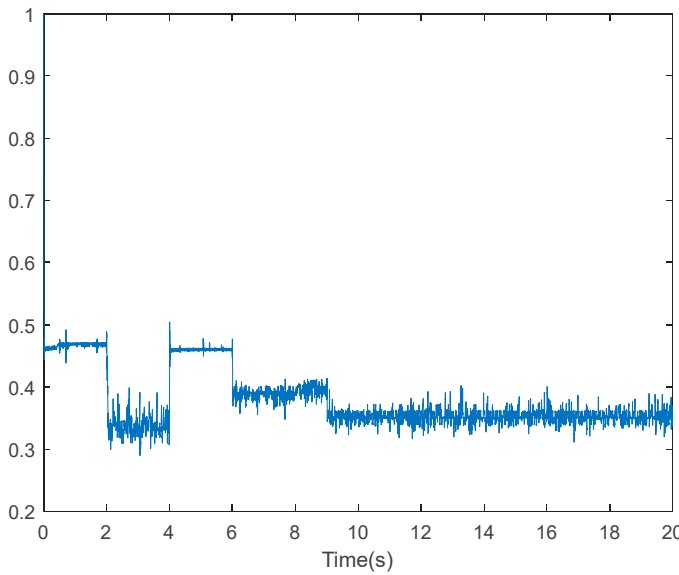

**Figure 17.** Controller signal.

Figures 17 and 18 show the controller signal and current error. This signal represents the duty cycle of the SEPIC converter which is limited between 0 and 1.

The previous model is applied to predict the temperature and the irradiation, while the found responses are depicted in the following figures.

In accordance with Figures 19 and 20, showing the temporal evolution of the DC motor current and voltage, respectively, Figure 20 represents the power of such a motor. The power is less than the PV generator power, since at the SEPIC output, the current is oscillating: the load (plus its filtering capacitor) is only supplied for a fraction of the cycle. This requires the use of large filter elements, which are, moreover, highly stressed. Figure 21 represents the speed of the DC motor. As the voltage and current change, the speed changes as well (See Figure 22).

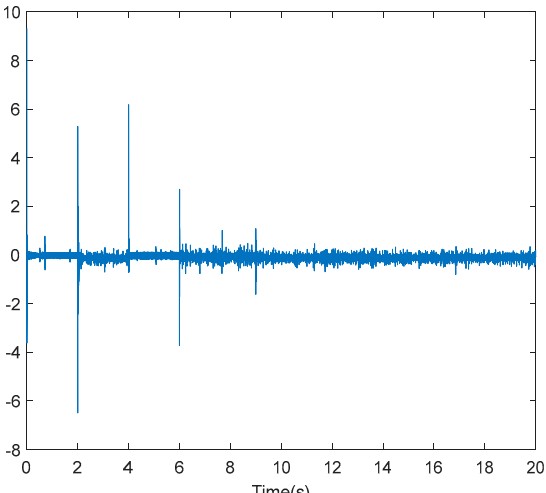

**Figure 18.** Current error.

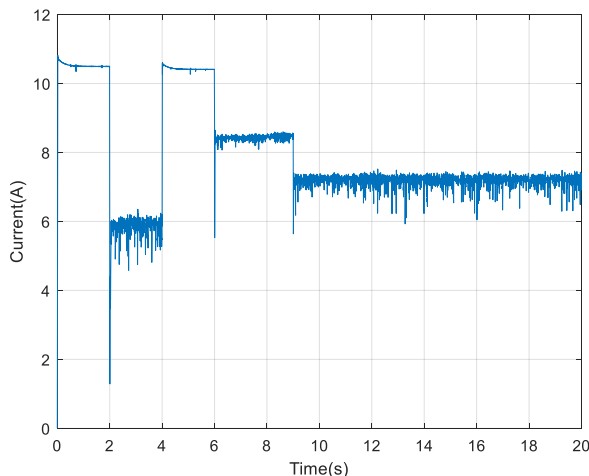

**Figure 19.** Temporal evolution of the motor current.

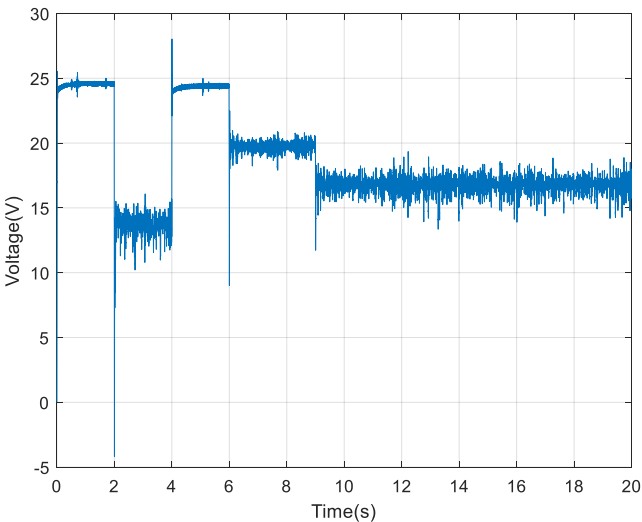

**Figure 20.** Temporal evolution of the motor voltage.

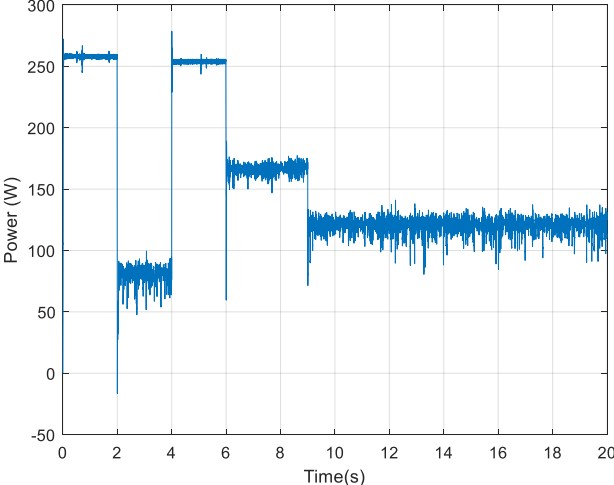

**Figure 21.** Temporal evolution of the motor power.

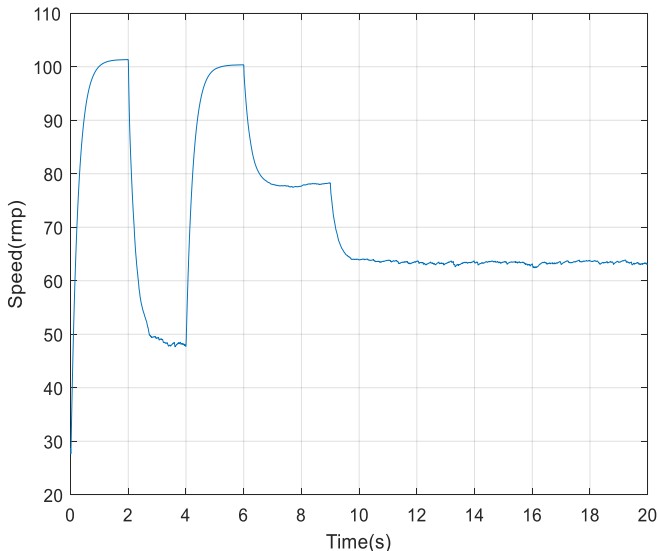

**Figure 22.** Temporal evolution of the motor speed.

The previous figures show the various changes in the DC motor behavior according to the predicted temperature and irradiation profiles. These changes are due to the MPP location changing. When the predictor reaches the predicted value, the algorithm finds the MPP point, so the curves join the steady state.

In Figures 23–25 at certain moments, the current, voltage, and power exceed the maximum value, which indicates an increase or decrease. This is due to the change in radiation and temperature at these moments, which indicates that the power has not yet reached the maximum value. The response stabilizes at a maximum of 2.5 ms, which is a reasonable value.

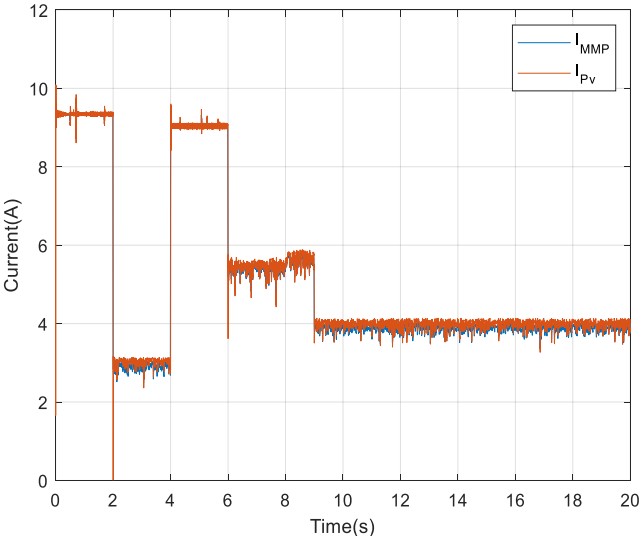

**Figure 23.** Temporal evolution of PV current over time.

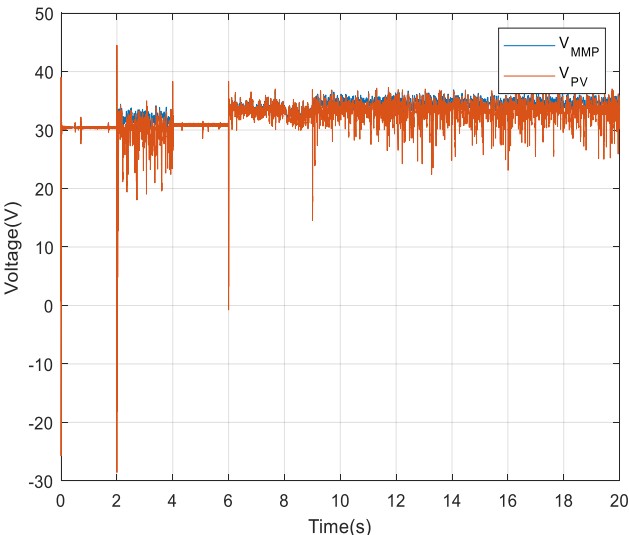

**Figure 24.** Temporal evolution of PV voltage over time.

The analysis of the dynamic behavior of the current, voltage, and generated power-highlights the success to follow the MPP point and, therefore, confirm the effectiveness of the proposed JAYA-SMC to find the maximum available power.

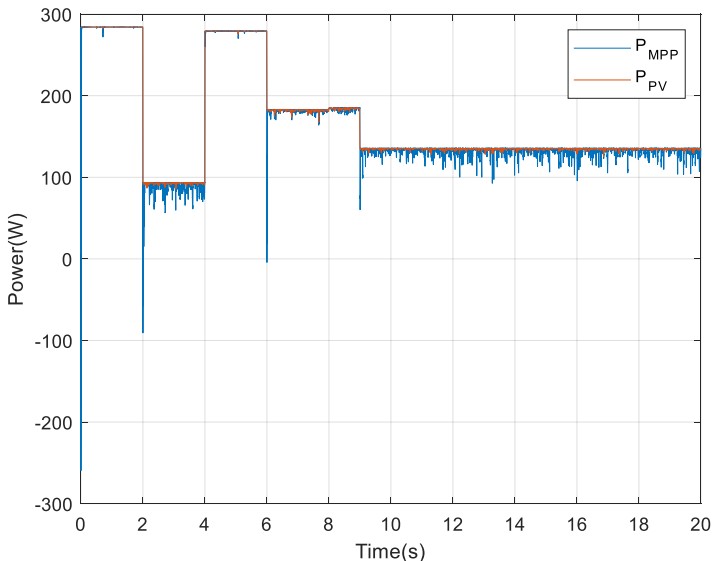

**Figure 25.** Temporal evolution of the PV generator power.

Following the presentation and discussion of the main achieved results, Table 3 briefly shows a comparative analysis that has been developed. This comparative study has focused on works that have dealt with the subject of prediction using different feasible approaches. The comparison is based on the studied system and has included the use of the prediction model, the main objective of the study, the degree of complexity of the concerned approach, and other performance and accuracy metrics. This comparative study highlights the competitiveness of the proposed method by benchmarking it against recently published techniques.

**Table 3.** Comparative study.

| Ref. | Studied System | Model | Main Objective | Degree of Complexity | MSE | R² |
|------|----------------|-------|----------------|---------------------|-----|-----|
| [51] | PVS | QSVM | Short-term energy forecasting for building integrated PV system | High | 0.16 | 0.88 |
| | PVS | Decision Tree | | High | 0.087 | 0.88 |
| [52] | Solar radiation-based power plants | CNN-BiLSTM | Midterm solar radiation prediction | High | 0.17 | 0.94 |
| [53] | Meteorological ground stations | DL | Estimation of daily solar radiation | High | 0.6 | 0.98 |
| [54] | Meteorologicalstation | ANFIS | Predict solar radiation | Low | 1.16 | 0.85 |
| [55] | SES | GMDH | Estimation of daily global solar radiation | Medium | 0.05 | 0.98 |
| [56] | SES | ANFIS-PSO | Monthly solar radiation prediction | High | 0.09 | 0.99 |

The ANN proposed model has been compared to the available literature: QSVM, Decision Tree [51], CNN-BiLSTM [52], DL [53], ANFIS [54], GMDH [55], and ANFIS-PSO [56] that have been proposed for many varieties of solar radiation prediction. The MSE and $R^2$ criteria were used for an accurate comparison. According to Table 3, the results clearly showed that the ANN model performs more accurately than the mentioned models according to the values of MSE = 0.003, and $R^2$ = 0.98. In addition, these studies are difficult to implement due to the issues related to the identification of hyper-parameters and specific parameters, which is not the case for the proposed approach.

## 4. Conclusions and Future Works

In this paper, we propose the application of artificial neurons and JAYA-SMC to predict, control, and research the maximum power of photovoltaic panels. For predicting the temperature and solar radiation on the PVS, artificial neural networks have been used. It is worth noting that this method is considered to be the most performant in this field due to its effectiveness. This is because it has hidden layers that provide it with a competitive advantage when it comes to predictive analysis. As a consequence, it is more accurate. As opposed to deep learning, it does not require external instructions when an error occurs or when an incorrect prediction is made. A mean square error of 0.03% and a regression coefficient of 0.98 were achieved with the proposed method. Based on these results, we can see that the target and the real output have a high degree of convergence and correlation. After that, JAYA-SMC was used to search and control the maximum power of the PVS by searching the MPP based on JAYA's algorithm. Additionally, the proposed method eliminates the inconveniences associated with OCV and SCC techniques, as well as the necessity of additional fixed keys and additional measurements. Since photovoltaic systems are dependent on converters so they can be connected to loads, the SEPIC duty cycle was controlled by the SMC by controlling the input current and using the Imp provided by JAYA as a reference. Simulation results demonstrate the effectiveness of the SMC. The current and voltage curves of the MPPT controller and PV are shown to be matched, indicating the controller's effectiveness.

In spite of the highly developed advanced forecasting method used in this work, the results still need to be improved. This is since this method has not been experimentally tested and has only been evaluated through simulation tests. There are several limitations associated with this method, such as storage and diagnosis, especially when the system is connected to other sources of energy. Due to advances in artificial intelligence, it can be used to diagnose and predict energy consumption, as well as to solve the storage problem faced by renewable energy sources.

Future work should include the integration of energy storage into the studied system to address the issue of PV solar source fluctuations. It is one of the most promising energy storage technologies to incorporate AEMFCs, since they combine the advantages of PEMFCs and the economics of alkaline batteries. Such an application would be a significant achievement that would enable the economic adoption of hydrogen energy storage systems throughout the world.

**Author Contributions:** O.B.: conceptualization, simulation, data collection, and review; A.K.-M.: simulation and data collection; M.J.: writing original draft; M.J., R.A., and H.J.: writing, review, and editing; F.H. and M.A.: supervision. All authors have read and agreed to the published version of the manuscript.

**Funding:** This study received no external funding.

**Data Availability Statement:** Not applicable.

**Acknowledgments:** The authors wish to express their gratitude to the Basque Government, through the project EKOHEGAZ (ELKARTEK KK-2021/00092), to the Diputación Foral de Álava (DFA), through the project CONAVANTER, and to the UPV/EHU, through the project GIU20/063, for supporting this work.

**Conflicts of Interest:** The authors declare no conflict of interest.

## Abbreviation List

| | |
|---|---|
| **ANN** | **Artificial Neural Network** |
| **AI** | Artificial intelligence |
| **NN** | Neural networks |
| **R** | Regression |
| **MSE** | Mean squared error |
| **MPP** | Maximum power point |
| **MPPT** | Maximum power point tracking |
| **P&O** | Perturb and observe |
| **Tanh** | Hyperbolictangent |
| **OCV** | Open-circuit voltage |
| **SCC** | Short-circuit current |
| **PVS** | Photovoltaic system |
| **SEPIC** | Single ended primary inductor converter |
| **SMC** | Sliding mode control |
| **mFFO** | Modified fire-fly optimizer |
| **FE-SVR** | Feature engineering-support vector regression |
| **FLC** | Fuzzy logic control |
| **GA** | Genetic algorithm |
| **PSO** | Particle swarm optimization |
| **CPSO** | Chaotic particle swarm optimization |
| **GWO** | Grey wolf optimization |
| **PI** | Proportional integral |
| **DC** | Direct current |
| **PID** | Proportional integral derivative |
| **PN** | Positive negative |
| **IC** | Incremental conductance |
| **SES** | Solar energy system |
| **QSVM** | Quadratic support vector machine |
| **CNN-BiLSTM** | Convolution neural network-bi-direction long short term memory |
| **ANFIS** | Adaptive neuron fuzzy inference system |
| **GMDH** | Group method of data handling |
| **ANFIS-PSO** | Adaptive neuron fuzzy inference system-particle swarm optimization |

## Appendix A

**Table A1.** PV panel parameters.

| Peak Power (Pmax) | 340 W |
|---|---|
| Voltage at Pmax (Vmp) | 36.7 V |
| Current at Pmax (Imp) | 9.28 A |
| Open circuit voltage (Voc) | 45.2 V |
| Short circuit current (Isc) | 9.9 A |

## Appendix B

**Table A2.** Converter parameters.

| Frequency PWM | 55 (KHz) | $L_1$ | 1.8 (mH) | $L_2$ | 1.4 (mH) |
|---|---|---|---|---|---|
| $C_1$ | 120 (μF) | $C_2$ | 470 (μF) | | |

## Appendix C

**Table A3.** DC motor parameters.

| Armature Resistance Ra | 2.2 Ω |
|---|---|
| Armature inductance La | $5 \times 10^{-3}$ H |
| Back-emf constant | 0.015 V/rmp |
| Total inertia J | 0.03 kg.m$^2$ |
| Viscous friction coefficient | 0.12 N.m.s |
| Coulomb friction torque Tf | 0.11 N.m |
| Initial speed | 3 rad/s |

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
