# Peer review of "An Artificial Neural Network for Solar Energy Prediction and Control Using Jaya-SMC"

_electronics, doi:10.3390/electronics12030592_

Round 1

Reviewer 1 Report

Please consult following paper for hyper-parameters optimization “Liquid-to-vapor phase change heat transfer evaluation and parameter sensitivity analysis of nanoporous surface coatings” and “On the Critical Heat Flux Assessment of Micro-and Nanoscale Roughened Surfaces”. In these works, Bayesian surrogate gaussian process, gradient boost regression trees and random forest were used for finding the optimal set of hyper-parameters.

Author Response

Please, see the atached file.

Reviewer 2 Report

The novelty of the work is not justified properly. 

Deep comparative analysis with existing technologies is required

Explanation on the achieved is satisfactory, it will make the readers interesting  

Author Response

Please, see the atached file.

Reviewer 3 Report

Comments:

This paper presents the design and implementation of artificial neural network for solar energy prediction and control using Jaya-SMC. The study is adequately described and supported by well-established results. However, the points of improvement are the following.

1. The Abstract section should be concise and informative and must be having flow. ‎Besides, please ‎provide ‎more ‎quantitative data in this section.

2. Typically the introduction comprises of four parts like motivation and background, related work, contribution, and organization. This manuscript lacks the typical structure.

3. This manuscript lacks the connection between artificial neural network, and Jaya-SMC. ‎Please add more explanations, create connection between artificial neural network, and Jaya-SMC, and give solid reasoning why artificial neural network, and Jaya-SMC.

4. Contributions of this work are not convincing, lack novelty, and need to be clarified.

Does not represent novelty and contribution. The contribution must be mentioned in second last paragraph of the induction in bullets form. 

5. The related work part of the manuscript is very rudimentary, please add more recent and relevant papers, and highlight the research gaps.

https://doi.org/10.1016/j.apenergy.2021.117178

6. Specify which data is used as a case study. The dataset used as a case study is very basic and does not convey any sense. There are other effective factors such as humidity, dry bulb, and temperature etc. conditions. Need major revision. 

7. There are equations with little explanation. This puts the entire burden on ‎the ‎reader ‎to figure out what is going on. It would be much more helpful if you would add ‎more ‎relevant ‎explanations for each equation. Further, all symbols should be defined in all ‎equations ‎, not some symbols in ‎several equations. Please explain clearly all symbols in the ‎formulas. ‎ Also, please define different indices throughout the formulation.

8. Please run the experiments for at least 20 times and record the best, mediocre, and ‎worst ‎results. Also, report the standard deviations.

‎9. Current limitations, future scope, scalability issues, and other useful ‎information ‎should be addressed in the Conclusion section.

Author Response

Please, see the atached file.

Reviewer 4 Report

In this work, a neural network-based technique is established, to forecast the operating temperature and radiation for the PVS. Besides, the meta-heuristic JAYA algorithm is implemented to track accurately the maximum power point (MPP) for a single-ended primary-inductor converter (SEPIC) supplying a DC motor. After reviewing carefully, the reviewer found that the paper has lots of scopes for further improvement.

1.       The abstract is poorly written. Rewrite the abstract by mentioning the advantages of the proposed method and numerical results at the end. 

2.       Why ANN has been used to predict the solar irradiation despite there are many new machine learning or deep learning algorithms are available.

3.       Add paper organization at the end of introduction.

4.       The quality of Figure 4 needs to be improved.

5.       There is no flow chart provided regarding how ANN is predicting solar irradiation.

6.       For prediction a big data set is required for solar irradiation. The author did not mention from where they got the data set.

7.       There is no comparison found to prove how the proposed ANN based prediction method is better than modern deep learning or machine learning algorithms.

8.       There is no comparison found how propose Jaya-SMC is better than other existing controllers.

9.       There are lots of grammatical and spelling mistakes observed which need to be rectified. Extensive English editing is required.

10.   How the data processing is conducted is not mentioned.

11.   More results are needed to prove the effectiveness of the research.

12.   A comparative study is missing in the discussion section to prove the effectiveness of the proposed method compared to the existing approaches which are proposed from 2019-2022.

13.   There is no abbreviation list provided in the article.

14.   The conclusion is also poorly written. No quantitative results are found to make a decision on the research work.

Author Response

Please, see the atached file.

Round 2

Reviewer 1 Report

Accepted

Author Response

The authors would like to thank the Reviewer 1 for his precious time and positive feedback on our work.  

Reviewer 2 Report

Authors have answered all the questions,  paper may be accepted in its present form.

Author Response

The authors would like to thank the Reviewer 2 for his precious time and positive feedback on our work.

Reviewer 4 Report

There is no technical comparison found to prove how the proposed ANN based prediction method is better than modern deep learning or machine learning algorithms. The authors need to include a comparison based on the results obtained by the proposed technique with other techniques.

Author Response

Please, see the reply in the attached file.
